# Risks of Carotid Artery Stenosis and Atherosclerotic Cardiovascular Disease in Patients with Calcium Kidney Stone: Assessment of Systemic Inflammatory Biomarkers

**DOI:** 10.3390/jpm12101697

**Published:** 2022-10-11

**Authors:** Chan-Jung Liu, Hau-Chern Jan, Ho-Shiang Huang

**Affiliations:** 1Department of Urology, National Cheng Kung University Hospital, College of Medicine, National Cheng Kung University, Tainan 704, Taiwan; 2Division of Urology, Department of Surgery, National Cheng Kung University Hospital Dou-Liou Branch, Yunlin 640, Taiwan

**Keywords:** kidney stone, carotid artery stenosis, atherosclerotic cardiovascular disease, systemic inflammatory biomarkers

## Abstract

(1) Background: To assess the clinical significance of preoperative inflammatory biomarkers combined with atherosclerotic cardiovascular disease (ASCVD) risk score to evaluate carotid artery stenosis in patients with calcium kidney stones; (2) Methods: We conducted a prospective observational case-control study, enrolling 74 patients with calcium kidney stones and 66 age- and sex-matched healthy controls. We calculated the inflammatory biomarkers including the neutrophil-lymphocyte ratio (NLR), monocyte-lymphocyte ratio (MLR), and systemic inflammation response index (SIRI). An ultrasound of the carotid arteries was performed on all participants to identify the severity of the stenosis; (3) Results: All inflammatory biomarkers and the severity of carotid artery stenosis were higher in the calcium kidney stone group than in controls. After stratification of ASCVD, inflammatory biomarkers and carotid artery stenosis severity were still significantly higher in the calcium kidney stone group. Multivariate analyses showed that calcium kidney stones significantly increased the risk of ASCVD and carotid artery stenosis. In multivariate linear logistic regression analyses, calcium kidney stone and ASCVD score had a significant association with carotid artery occlusion, but SIRI did not; (4) Conclusions: Calcium kidney stone is associated with higher levels of inflammatory biomarkers and carotid artery stenosis. Calcium kidney stone is associated with higher levels of inflammatory biomarkers and carotid artery stenosis.

## 1. Introduction

Kidney stones are one of the most common urologic diseases, with increasing incidence and prevalence worldwide [1]. More than 80% of kidney stones are calcium-containing, comprising calcium oxalate and phosphate, either alone or combined [2]. Calcium kidney stones have numerous etiologic factors, including genetic, dietary, metabolic, and infective factors [3]. However, most calcium stone formers are idiopathic, with no noticeable metabolic abnormalities [4]; recently, various studies have found these formers to be associated with a higher risk of developing atherosclerotic cardiovascular disease (ASCVD) and carotid artery atherosclerosis [5]. Although some authors believe that the pathophysiology underlying this association could be lifestyle and metabolic syndromes, the exact mechanism is not fully understood. The characterization of the pathogenesis of kidney stone formation has been challenging, and providing strong evidence to establish the link between ASCVD, carotid artery stenosis, and calcium kidney stones, is a subject of ongoing research.

Inflammation and oxidative stress facilitate kidney crystal adhesion to renal tubular cells and crystal formation, leading to kidney stone growth [6,7], and inflammation contributes to the progression of coronary and carotid arteriosclerotic diseases [8]. Given that inflammation plays a key role in both kidney stones and carotid artery stenosis, many inflammatory indicators, including neutrophil-lymphocyte ratio (NLR) and monocyte-lymphocyte ratio (MLR), can be combined to function as biomarkers. The systemic inflammation response index (SIRI), which integrates different inflammatory cells (neutrophils, monocytes, and lymphocytes), has been considered a more accurate inflammatory biomarker and has been proven to predict the prognosis of patients with different cancers [9]. Considering the growing evidence on the relationship between calcium kidney stones, ASCVD, and carotid artery stenosis, we examined whether their levels are associated with the presence of calcium kidney stones and the severity of carotid artery stenosis. We also sought to determine whether the severity of carotid artery stenosis worsened in calcium stone formers compared to non-stone formers, stratified according to ASCVD risk scores.

## 2. Materials and Methods

### 2.1. Study Design and Population

This was a retrospective, observational, cohort study, and the study protocols involving human materials were approved by the Cheng Kung University Hospital institutional ethics committee (IRB: B-ER-103-400 and A-ER-108-425). We enrolled calcium-containing kidney stone formers from urology clinics at a tertiary medical center between July 2017 and September 2019. All patients with a confirmed diagnosis of calcium kidney stones were included, based on the stone analysis results from treatment with extracorporeal shock wave lithotripsy, retrograde intrarenal surgery, or percutaneous nephrolithotomy, at least 6 months before enrollment. The exclusion criteria were as follows: (1) age < 18 years; (2) those who had been or were currently undergoing statin treatment prior to the study; (3) those who were currently taking stone-controlling medications including allopurinol, febuxostat, benzbromarone, potassium citrate, and thiazides; and (4) those with stone components other than calcium oxalate or phosphate. Age- and sex-matched non–stone formers were drawn from urology clinics at the same tertiary medical center. These patients underwent thorough clinical evaluation by a urologist, including renal sonography, intravenous pyelography, and noncontrast computed tomography of the abdomen and pelvis. The definition of non-stone formers was based on a previously published study [5,10]. All participants signed an informed consent form for participation in the study. All methods were performed in accordance with the relevant guidelines and regulations.

### 2.2. Data Collection

For eligible participants, demographic and clinical characteristics including age, gender, body height, bodyweight, abdominal circumference, serum lipid profiles, and comorbidities, were collected. The 24 h urine collection was analyzed for creatinine, calcium, phosphorus, magnesium, and uric acid levels. Complete blood count (CBC) parameters were recorded in all participants to determine the NLR, MLR, and SIRI. The following exclusion criteria were applied to potential participants: (1) active infection status; and (2) absence of differential count information from preoperative CBCs 30 days before enrollment, which is the same as previously published [11]. Data above were collected retrospectively from patient health records at calcium nephrolithiasis diagnosis and at the first follow-up visit after enrolling this cohort. Data from non–stone formers were collected retrospectively from health records after excluding the diagnosis of nephrolithiasis and at the first follow-up visit after enrolling this cohort. After enrolling this cohort, carotid ultrasonography was assigned to all participants. The 10-year ASCVD risk score was calculated using the Pooled Cohort Equations for other races, including age, sex, total cholesterol level, HDL cholesterol, systolic blood pressure (BP) (using different coefficients, depending on whether or not the individual was treated for hypertension), current smoking status, and the presence or absence of diabetes mellitus [12]. Systolic BP values were obtained at clinics. The calculation of this score was made on the “ASCVD Risk Estimator” website. This allowed us to stratify the study participants into three groups: low 10-year ASCVD risk (<7.5%), intermediate 10-year ASCVD risk (≥7.5% to <20%), and high 10-year ASCVD risk (≥20%) groups.

### 2.3. Assessment of Carotid Ultrasonography

An extracranial carotid artery doppler sonography was performed by well-trained, protocol-adherent technicians at the Department of Neurology. The severity of carotid atherosclerosis in each subject was evaluated using two parameters: intima-media thickness (IMT); and maximum percentage stenosis for IMT and carotid plaques. IMT was measured as the distance between the leading edges of the lumen-intima and media-adventitia interfaces on the far wall of the common carotid artery (CCA) and the internal carotid artery (ICA). The maximum IMT was the greatest thickness of the wall, including plaque lesions. The maximum percentage of stenosis was calculated by measuring the residual lumen diameter and the original diameter at the site of maximal stenosis and dividing the difference by the original diameter.

### 2.4. Statistical Analysis

Continuous variables were compared between the two groups using Student’s *t*-test, and the nominal and categorical variables were compared using the chi-squared likelihood ratio or Fisher’s exact test, rejecting the null hypothesis at *p* < 0.05. All data are expressed as mean ± SD. We used a multiple linear regression model to compare SIRI, the presence of calcium kidney stones, and the ASCVD risk score to predict the severity of carotid artery stenosis. The correlation between the ASCVD risk score and carotid artery stenosis was evaluated using Spearman’s rank correlation coefficient. We determined ROC curves, generated by computing sensitivity and specificity, calculated the area under the curve including 95% confidence intervals, and compared these areas using nonparametric tests. Comparisons between the three groups were performed using 1-way ANOVA followed by Tukey’s multiple comparisons test. Statistical analyses were performed using the SPSS Statistics (V20; IBM Corp, Armonk, NY, USA).

## 3. Results

### 3.1. Comparison of Clinical Baseline, Serum Inflammation Biomarkers, 24 h Urinary Biochemistry, Lipid Metabolism Parameters, and Cardiovascular Risk Parameters in Participants with and without Calcium Kidney Stones

A total of 140 patients were enrolled in this prospective study. Of them, 74 were calcium kidney stone formers, while 66 were stone-naïve controls (Table 1). There were no significant differences in age, BMI, abdominal circumference, and sex distribution, between the groups. More than 60% of the patients in both groups were men. We did not find any significant differences in all 24 h urine chemistries. Among all calcium kidney stone formers, 32 were hypertensive (32%), 46 had diabetes mellitus (33%), 41 had high TG (29%), 97 had high cholesterol (69%), 107 had elevated LDLs (76%), and 31 had reduced HDLs (22%). Compared with the controls, there were significant differences in high TG (*p* = 0.006), high LDL (*p* = 0.009), and reduced HDL (*p* < 0.001). In addition, pre-treatment inflammation markers including NLR (*p* = 0.002), MLR (*p* < 0.001), and SIRI (*p* = 0.001), were all significantly higher in stone formers. Max-IMT and carotid artery stenosis percentage were both significantly higher in stone formers than in controls. According to the different severities of carotid artery stenosis, the stenosis percentage was relatively more severe in calcium kidney stone formers than in controls (*p* = 0.003). Among calcium kidney stone formers, <7.5%, 7.5–19.9%, and >20% ASCVD risks, were observed in 41 (55%), 26 (35%), and 7 (10%) individuals, respectively. The distribution of ASCVD risk scores appeared to be insignificant between the groups, but there was a significant trend (*p* = 0.076).

### 3.2. Relationship between Calcium Kidney Stone, Cardiovascular Disease Risk, and Carotid Artery Atherosclerosis

First, a scatter plot and linear regression analysis of the ASCVD risk score and carotid artery stenosis demonstrated that the ASCVD risk score was associated with an increase in the severity of carotid artery stenosis in both calcium kidney stone formers and stone-naive controls (*p* < 0.05) (Figure 1). The association was more significant in the calcium kidney stone group than in the control group (r^2^ = 0.150 vs. r^2 =^ 0.032). In this regression model, a significant overall F-test (F value = 13.024, *p* < 0.05) and an individual *t*-test (t value = 3.609, *p* < 0.05) should also be emphasized.

Second, we further evaluated the association between calcium kidney stones and carotid artery atherosclerosis in different ASCVD risk score populations. Initially, all 140 participants were divided into three groups based on the ASCVD risk scores (Table 2). For comparison of Max-IMT between calcium kidney stones and controls in the three ASCVD risk score groups, larger Max-IMT apparently corresponded to a higher ASCVD risk score, and calcium kidney stone formers had a markedly larger Max-IMT value than the controls (*p* < 0.001). Max-IMT in calcium kidney stone formers and controls was 0.89 ± 0.32 and 0.71 ± 0.09, respectively, in the high-risk group; 0.76 ± 0.11 and 0.70 ± 0.00, respectively, in the intermediate-risk group; and 0.61 ± 0.12 and 0.60 ± 0.10, respectively, in the low-risk group. Regarding the severity of carotid artery stenosis, the presence of calcium kidney stones was associated with a higher risk of carotid artery stenosis (*p* = 0.003) (Figure 2). In the high-risk group, 67% of the controls and 71% of calcium kidney stone formers had 25 –35% and more than 35% occlusion of the carotid artery, respectively. However, in the low-risk group, only 42% of calcium kidney stone formers and 27% of controls had more than 25% stenosis of the carotid artery. In addition, the values of pre-treatment inflammation markers (NLR, MLR, and SIRI) were significantly higher in the high-risk group than in the intermediate- and low-risk groups, but there was no significant difference between the kidney stone and control groups, regardless of the ASCVD risk (Table 2).

Using logistic regression analysis, we evaluated the risk factors for high ASCVD risk score and carotid artery stenosis (Table 3). First, age, the presence of calcium kidney stones, BMI, and more than 25% carotid artery stenosis, were significantly associated with increased risks of ASCVD on univariate analysis, whereas only age, calcium kidney stone, and BMI were associated with higher ASCVD risks on multivariable analysis. Instead, age, male sex, presence of calcium kidney stones, and higher ASCVD risk scores, were significantly associated with higher risks of carotid artery stenosis on univariate analysis, but only age, male sex, and calcium kidney stones were associated with deteriorated carotid artery stenosis.

### 3.3. Combination of Calcium Kidney Stone and ASCVD Risk Scores as a Predictive Index of Carotid Artery Stenosis

This study confirmed the correlation between calcium kidney stones and the ASCVD risk scores with carotid artery occlusion. In the multivariate logistic regression model, age and calcium kidney stones were related to elevated ASCVD and severe carotid artery stenosis. In multivariate linear logistic regression analyses, calcium kidney stone (β = 0.295, *p* < 0.001) and the ASCVD risk scores (β = 0.287, *p* < 0.001) were significantly associated with carotid artery occlusion, but the inflammation marker SIRI was not (Table 4). Furthermore, the ROC analysis was performed to assess the ability of the combination of calcium kidney stone and the ASCVD risk scores in predicting carotid stenosis, which showed an area under the ROC curve (AUC) = 0.677 (95% CI: 0.587–0.786) and *p* < 0.001 (Figure 3).

## 4. Discussion

Our results show that calcium kidney stone formers were associated with a higher risk of having high TG and low HDL, compared with age-, BMI-, and sex-matched stone-naïve controls. A significant increase in NLR, MLR, and SIRI, was also observed in calcium kidney stone formers. Carotid artery stenosis was more severe in calcium kidney stone formers than in controls. Using multivariate logistic regression analysis, the combination of calcium kidney stones and ASCVD risk could predict the severity of carotid artery stenosis. To the best of our knowledge, this is the first study to describe an association between calcium kidney stones and serum inflammation biomarkers (NLR, MLR, and SIRI).

It is widely acknowledged that kidney stones are associated with several ASCVDs including myocardial infarction, stroke, and vascular calcification [12,13]. Atherosclerotic carotid artery stenosis is an important cause of ischemic stroke and is strongly associated with vascular calcification [14]. Over a decade ago, a large cohort, named Coronary Artery Risk Development in Young Adults (CARDIA), demonstrated that young adults with a history of kidney stones were associated with silent carotid atherosclerosis [15]. This remarkable finding provides strong evidence that kidney stones may be an early indicator of carotid artery stenosis. Population-based studies have found that patients with nephrolithiasis have an increased risk of ischemic stroke development, which is consistent with the findings of early carotid atherosclerosis [16]. We first used different populations with specific stone compositions to analyze the association between carotid IMT and urinary stone risk factors [5]. Only calcium-containing stone formers had a significantly thicker carotid IMT than stone-naïve controls. However, these findings were also challenged by selection bias, as these enrolled kidney stone patients may already have a higher risk of atherosclerosis. To consider the well-known association between atherosclerosis and carotid artery stenosis, in this study, we tried to demonstrate that the association between calcium urolithiasis and carotid artery stenosis could still exist independent of atherosclerosis. We used the ASCVD risk score, which is a reliable scoring modality, to evaluate the risk of atherosclerosis, and to re-evaluate whether the presence or absence of calcium kidney stones would increase the risk of carotid artery stenosis. After adjusting for the ASCVD risk score, the severity of carotid artery stenosis was still increased in calcium kidney stone formers (Table 2). Multivariate regression analysis revealed that the presence of calcium kidney stones significantly increased the risk of carotid artery stenosis (OR = 3.66, *p* = 0.008). However, the association between calcium kidney stones and carotid artery stenosis may operate in part through multiple shared pathogenic mechanisms. Various comorbidities such as hypertension, DM, and chronic kidney disease, increase the risk of these two diseases [17,18]. Western dietary style and obesity are also possible links between calcium kidney stones and carotid artery stenosis [19]. Dysregulation of calcium homeostasis and the dysfunction of calcium-sensing receptors may be the possible causes of these two distinct diseases [19].

Inflammation has been recognized as a localized protective reaction of tissue to injury or infection; furthermore, recent studies have revealed its role in a wide variety of chronic diseases including cardiovascular disease, cancer, and diabetes mellitus. Accumulating evidence has reported that NLR, MLR, and SIRI, are prognostic indicators in several malignancies [20]. Given that systemic inflammatory responses participate in multiple diseases other than cancer, a growing body of literature has proposed that NLR, MLR, and SIRI, can also be potential predictors of various diseases, including peripheral arterial disease (PAD) [21], mortality after coronary artery bypass surgery [22], outcomes of target vessel restenosis after infrainguinal angioplasty [23], and end stage renal disease (ESRD) [24,25]. However, very few studies have focused on the role of systemic inflammatory responses in the formation of kidney stones [26]. Considering that inflammation promotes kidney stone formation, NLR, MLR, and SIRI, are anticipated to be significant predictors of kidney stone disease [27]. In this study, we found a significant increase in NLR, MLR, and SIRI, in calcium stone formers compared to controls. After stratification of the ASCVD risk score, NLR, MLR, and SIRI, continued to increase in calcium kidney stone formers (Table 2). The mechanisms responsible for the clinical significance of the association of these inflammation indices with calcium kidney stones might be explained by the functions of neutrophils, lymphocytes, and monocytes. First, renal crystal-induced inflammation activates the NLRP3 inflammasome and triggers caspase-1-dependent secretion of IL-1β and IL-18 secretion, which induces the recruitment of neutrophils and macrophages [28]. Second, monocytes, the precursors of macrophages, can differentiate into two types of macrophages (M1 and M2) to modulate inflammation. Renal crystal development is facilitated by M1 but inhibited by M2 [29]. Hence, monocytes play a fundamental role in kidney stone formation. Lymphocytes are mainly involved in adaptive immunity and are strongly associated with acute kidney injury in both human and animal studies [30]. Kidney stone is one of the main causes of acute kidney injury. Therefore, lymphocytes may be involved in kidney stones. It is worth mentioning that only a high NLR (≥2.1) was associated with an increased risk of calcium kidney stones using multivariate logistic regression analysis in this study (OR = 2.77, *p* = 0.015). Mao et al. also demonstrated this significant finding using their cohort [26]. A possible explanation for this specific result is that existing strong evidence of NLRP3 inflammasome participation in kidney inflammation caused by kidney stones, which is closely related to neutrophils [31,32].

Recent studies have suggested that metabolic syndrome increases the risk of kidney stones [33]. Lifestyle, dietary habits, and acidic urine may explain the increased risk of kidney stones. Participants with metabolic syndrome are at increased risk for both progressive carotid atherosclerosis and ASCVD [34]. Past studies have only demonstrated an association between kidney stones and atherosclerosis. In this study, the results from multiple linear regression and ROC analyses suggest that the presence of calcium kidney stones is likely to be an exacerbating factor of carotid atherosclerosis among patients with existing risks of ASCVD. We already know that most calcium stones are idiopathic [4]. Some obscure pathophysiological differences are supposed to exist in calcium stone formers compared with lifelong non-stone formers. These possible differences include altered calcium signaling and oxidized low-density lipoprotein metabolism, and these physiological characteristics could all lead to carotid artery stenosis [13,35,36]. However, none of these theories is generally accepted. Hence, this unique and interesting finding from our study may be relevant and requires further study.

Nonetheless, our study had some limitations. First, it was retrospective in nature, and possible selection, detection, and performance of analysis bias, might be confounded. Second, the data were collected from a single institution. Moreover, the study lacked ideal and generalizable thresholds for blood-based parameters in kidney stone disease. Therefore, our results should be validated in different populations and ethnic groups. Additional research will advance our understanding of the use of inflammatory biomarkers in kidney stone disease.

## 5. Conclusions

We demonstrated that calcium kidney stones were associated with higher levels of inflammatory biomarkers including NLR, MLR, and SIRI, compared with age- and sex-matched non-stone formers. The severity of carotid artery stenosis was also higher among calcium stone formers than non-stone formers. After the stratification of the ASCVD scores, all inflammatory biomarkers and the severity of carotid artery stenosis were still significantly higher in calcium stone formers. Using multivariate logistic regression analysis, the presence of calcium kidney stones is a risk factor for both ASCVD and carotid artery stenosis. Given that inflammation plays a key role in kidney stones and atherosclerosis, further studies utilizing serum inflammation biomarkers to unveil the link between calcium kidney stones, carotid artery stenosis, and ASCVD, are warranted. We recommend that clinicians should be aware of the possible cardiovascular comorbidities in patients with calcium kidney stones.

## Figures and Tables

**Figure 1 jpm-12-01697-f001:**
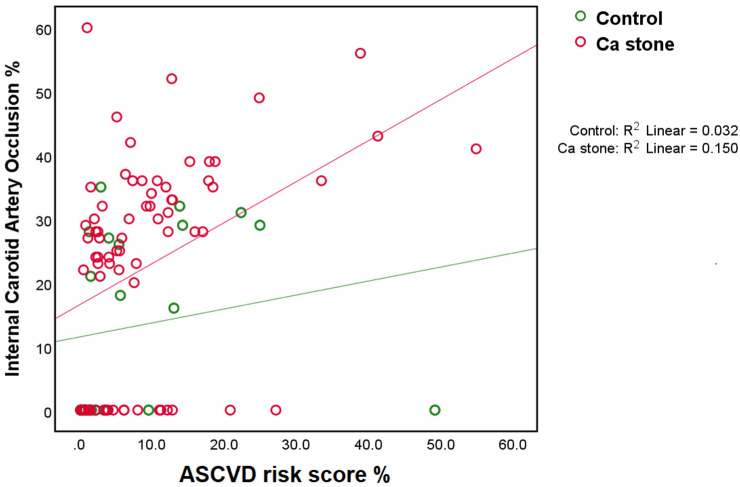
Scatter plot of ASCVD risk score versus carotid artery stenosis (%) with linear regression. The square of the correlation coefficient (r2) with significance (*p* value) is presented in the box.

**Figure 2 jpm-12-01697-f002:**
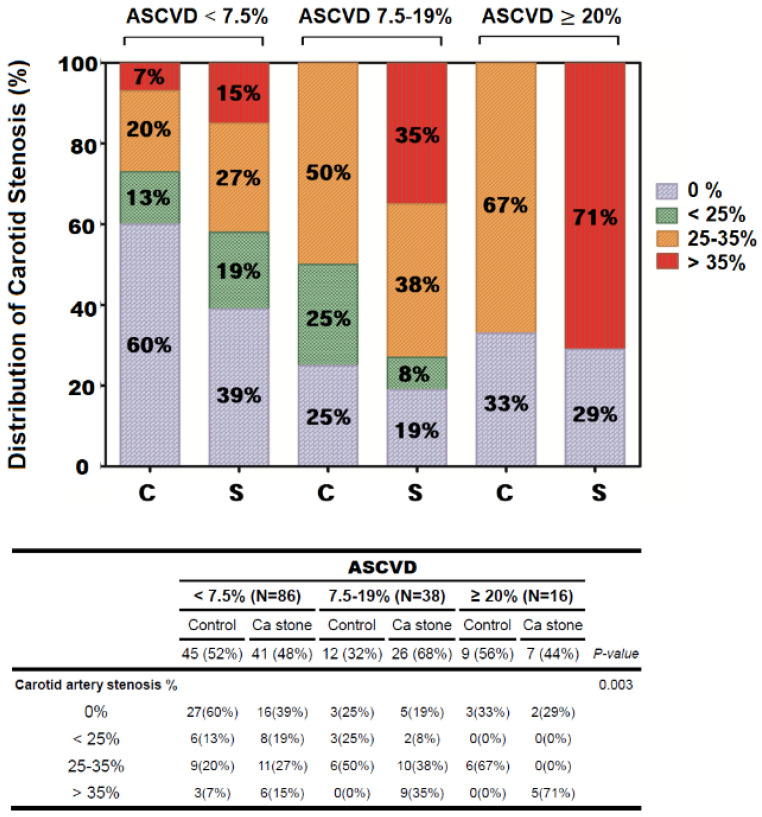
A bar chart for comparing distribution of carotid artery stenosis (%) between the control and kidney stone groups according to three different ASCVD risk scores. FRS: Framingham risk score, C: control, S: kidney stone.

**Figure 3 jpm-12-01697-f003:**
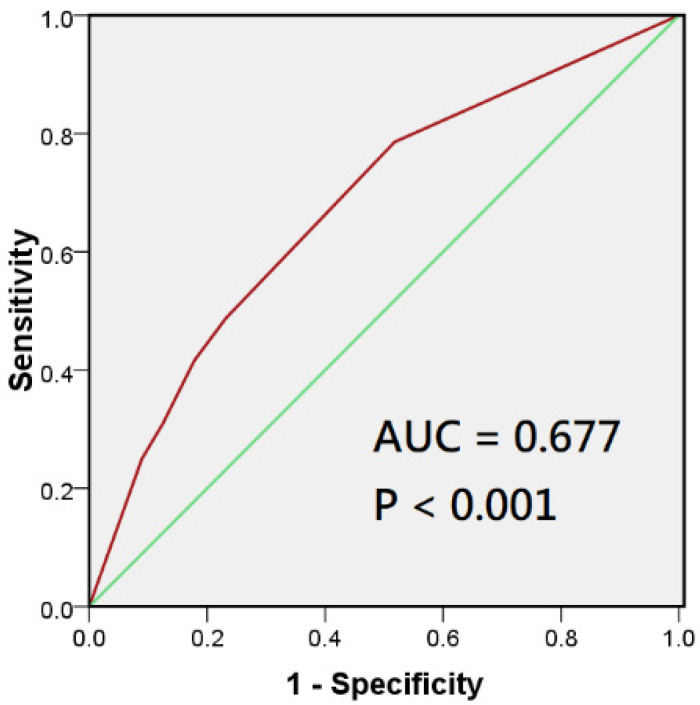
Receiver operating characteristic (ROC) analysis of the sensitivity and specificity of the prediction of carotid artery stenosis based on ASCVD risk score and calcium kidney stone.

**Table 1 jpm-12-01697-t001:** Comparison of basic data, 24 h urinary biochemistry, pre-treatment serum inflammation markers, and cardiovascular risk between patients with and without calcium-containing stone. Data presented as mean ± SD.

Variables	Ca Kidney Stone	Control	*p* Value
	(*n* = 74)	(*n* = 66)	
Age (y)	54.3 ± 10.3	53.4 ± 11.4	0.618
BMI (kg/m^2^)	26.1 ± 4.1	25.7 ± 3.6	0.562
Male (N, %)	47, 64	42, 64	0.988
Comorbidities			
Diabetes mellitus (N, %)	25, 34	21, 32	0.805
Hypertension (N, %)	27, 36	18, 27	0.224
High TG level (N, %)	29, 39	12, 18	0.006
High cholesterol level (N, %)	46, 62	51, 77	0.053
High LDL level (N, %)	50, 68	57, 86	0.009
Reduced HDL-C (N, %)	25, 34	6, 9	<0.001
24-h Urinalysis			
pH	6.0 ± 0.6	6.0 ± 0.7	0.998
P (mg/day)	771.9 ± 465.2	733.1 ± 257.2	0.55
Ca (mg/day)	226.3 ± 142.1	199.5 ± 93.1	0.195
Mg (mg/day)	101.1 ± 47.5	99.1 ± 33.9	0.083
Uric acid (mg/day)	666.8 ± 257.5	673.3 ± 218.9	0.873
Ccr (mL/min)	107.6 ± 51.8	105.2 ± 36.4	0.757
Pre-treatment peripheral blood leukocyte parameters			
NLR	4.4 ± 3.6	2.9 ± 1.5	0.002
MLR	0.4 ± 0.3	0.3 ± 0.1	<0.001
SIRI	2.5 ± 2.4	1.4 ± 0.9	0.001
Carotid artery ultrasonographic parameters			
Max-IMT (mm)	0.7 ± 0.2	0.6 ± 0.1	0.027
Stenosis %	22.6 ± 17.0	13.3 ± 13.1	0.001
Presence of stenosis (N, %)	51, 69	33, 50	0.023
Severity of carotid stenosis (N, %)			0.003
0%	23 (31%)	33 (50%)	
<25%	10 (14%)	9 (14%)	
25–35%	21 (28%)	21 (32%)	
>35%	20 (27%)	3 (4%)	
ASCVD risk score (%)			0.076
<7.5%	41 (55%)	45 (68%)	
7.5–19.9%	26 (35%)	12 (18%)	
≥20%	7 (10%)	9 (14%)	

Abbreviations: BMI, body mass index; TG, triglyceride; LDL, low-density lipoprotein; HDL-C, high-density lipoprotein cholesterol; 24-h, 24-h; P, Phosphate; Ca, calcium; Mg, Magnesium; Ccr, Creatinine clearance rate; NLR, Neutrophil-to-lymphocyte ratio; MLR, Monocyte-to-lymphocyte ratio; SIRI, systemic inflammation response index (neutrophil × monocyte/lymphocyte); Max-IMT, the maximal value among the measurements of the bilateral intima-media thickness of the carotid artery; Stenosis%, maximum percentage stenosis of carotid artery; ASCVD, atherosclerotic cardiovascular disease.

**Table 2 jpm-12-01697-t002:** Comparison of basic data, 24 h urinary biochemistry, and pre-treatment serum inflammation markers between patients with and without calcium-containing stone in different ASCVD risk score. Data presented as mean ± SD.

	ASCVD Risk Score						
	<7.5% (n = 86)		7.5–19.9% (n = 38)		≥20% (n = 16)		*p* Value
	Ca Stone	Control	Ca Stone	Control	Ca Stone	Control	
	41 (48%)	45 (52%)	26 (68%)	12 (32%)	7 (87%)	9 (13%)	
Age (y)	48.6 ± 8.7	48.5 ± 9.2	59.2 ± 5.0	60.0 ± 7.4	70.0 ± 7.6	69.3 ± 5.1	0.618
BMI (kg/m^2^)	25.8 ± 4.6	24.8 ± 4.0	26.3 ± 3.6	27.2 ± 5.0	26.9 ± 2.5	28.4 ± 1.1	0.562
Large Abd. Circ. (N, %)	25 (61%)	24 (53%)	20 (77%)	9 (75%)	7 (100%)	9 (100%)	0.404
Male (%)	24 (59%)	27 (60%)	19 (73%)	12 (100%)	4 (57%)	3 (33%)	0.988
Comorbidity							
Diabetes Mellitus (N, %)	9 (22%)	6 (13%)	12 (46%)	6 (50%)	4 (57%)	9 (100%)	0.805
Hypertension (N, %)	6 (15%)	6 (13%)	17 (65%)	6 (50%)	4 (57%)	6 (67%)	0.244
High TG level (N, %)	17 (42%) *	3 (7%)	11 (42%)	6 (50%)	1 (14%)	3 (33%)	0.006
High Cholesterol level (N, %)	27 (66%)	39 (87%)	14 (54%)	6 (50%)	5 (71%)	6 (67%)	0.053
High LDL level (N, %)	27 (66%) *	42 (94%)	17 (65%)	9 (75%)	6 (86%)	6 (67%)	0.009
Reduced HDL-C (N, %)	13 (32%) *	3 (7%)	10 (39%)	3 (25%)	2 (29%)	0 (0%)	0.001
24-h Urine Test							
pH	6.0 ± 0.7	6.2 ± 0.8	6.2 ± 0.7	5.8 ± 0.5	6.0 ± 0.0	6.0 ± 0.0	0.567
P (mg/day)	743.4 ± 311.4	733.1 ± 195.7	852.41 ± 750.31	646.80 ± 437.28	732.93 ± 277.88	819.87 ± 323.29	0.55
Ca (mg/day)	204.0 ± 131.2	182.0 ± 103.3	261.68 ± 157.18	247.81 ± 105.70	263.46 ± 67.92	240.47 ± 26.92	0.282
Mg (mg/day)	99.5 ± 44.2	94.3 ± 36.3	105.50 ± 60.08	106.13 ± 23.69	99.30 ± 30.53	117.5 ± 21.49	0.774
Uric acid (mg/day)	655.4 ± 273.2	651.2 ± 219.6	695.34 ± 254.75	644.57 ± 193.39	661.84 ± 160.40	820.23 ± 201.58	0.873
Ccr (mL/min)	118.9 ± 56.7	108.5 ± 35.3	89.82 ± 34.36	86.03 ± 43.92	82.63 ± 38.29	107.35 ± 32.47	0.757
Pre-Treatment Inflammation Markers							
NLR	4.2 ± 3.2	2.8 ± 1.6	4.9 ± 4.4	3.8 ± 1.2	3.8 ± 2.3	2.4 ± 0.6	0.002
MLR	0.4 ± 0.3	0.3 ± 0.1	0.5 ± 0.3	0.5 ± 0.2	0.4 ± 0.3	0.3 ± 0.1	<0.001
SIRI	2.4 ± 2.3	1.2 ± 0.4	2.6 ± 2.4	2.5 ± 1.5	2.9 ± 2.7	1.0 ± 0.3	0.001
Max-IMT (mm)	0.6 ± 0.1	0.6 ± 0.1	0.8 ± 0.1	0.7 ± 0.0	0.9 ± 0.3 *	0.7 ± 0.1	<0.001
Stenosis %	18.2 ± 16.3 *	10.3 ± 13.3	26.9 ± 14.6 *	19.3 ± 13.2	32.1 ± 22.8 *	20.0 ± 15.0	0.001
Presence of carotid artery stenosis (N, %)	25 (61%)	18 (40%)	21 (81%)	9 (75%)	5 (71%)	6 (67%)	0.023

* *p* < 0.05 vs. control. Abbreviations: BMI, body mass index; TG, triglyceride; LDL, low-density lipoprotein; HDL-C, high-density lipoprotein cholesterol; 24-h, 24-h; P, Phosphate; Ca, calcium; Mg, Magnesium; Ccr, Creatinine clearance Rate; NLR, Neutrophil-to-lymphocyte ratio; MLR, Monocyte-to-lymphocyte ratio; SIRI, systemic inflammation response index (neutrophil × monocyte/lymphocyte); Max-IMT, the maximal value among the measurements of the bilateral intima-media thickness of the carotid artery; Stenosis%, maximum percentage stenosis of the carotid artery.

**Table 3 jpm-12-01697-t003:** Logistic regression analysis for predictors of high ASCVD risk score and carotid artery stenosis.

	Univariate Analysis			Multivariate Analysis		
	OR	95% CI	*p* Value	OR	95% CI	*p* Value
	ASCVD Risk Score					
Age	1.23	1.15–1.31	<0.001	1.30	1.17–1.44	<0.001
Sex Female	Reference					
Sex (vs F)	1.64	0.81–3.33	0.17			
BMI	1.15	1.04–1.26	0.005	1.19	1.02–1.39	0.03
Cal. kidney stone	2.34	1.20–4.76	0.01	4.41	1.40–13.87	0.01
Lipid profiles						
Cholesterol	0.54	0.25–1.14	0.10			
HDL	1.01	0.43–2.36	0.99			
LDL	0.78	0.34–1.77	0.55			
Inflammatory biomarkers						
NLR	1.09	0.96–1.23	0.17			
SIRI	1.20	1.00–1.45	0.06			
MLR	6.67	1.31–33.86	0.02	3.65	0.30–43.83	0.31
Carotid artery stenosis						
No occlusion	Reference					
Occlusion < 25%	0.98	0.24–4.07	0.98	1.10	0.24–5.07	0.90
Occlusion 25–35%	4.75	1.86–12.10	0.001	0.74	0.18–2.97	0.67
Occlusion > 35%	6.79	2.28–20.19	<0.001	2.07	0.42–10.37	0.37
	Carotid artery stenosis					
Age	1.08	1.04–1.12	<0.001	1.22	1.13–1.32	<0.001
Sex Female	Reference					
Sex (vs F)	4.56	2.19–9.52	0.001	24.19	7.30–80.14	<0.001
BMI	1.04	0.95–1.13	0.43			
Cal. kidney stone	2.22	1.11–4.42	0.02	2.90	1.19–7.10	0.02
Lipid profiles						
Cholesterol	0.54	0.25–1.17	0.12			
HDL	1.07	0.47–2.43	0.87			
LDL	0.48	0.20–1.13	0.09			
Inflammatory biomarkers						
NLR	1.03	0.91–1.17	0.61			
SIRI	1.20	1.00–1.45	0.06			
MLR	2.30	0.48–10.96	0.30			
ASCVD risk scores	1.05	1.01–1.09	0.03	0.93	0.88–0.98	0.09

Abbreviations: ASCVD, atherosclerotic cardiovascular disease; OR, odds ratio; vs., versus; CI, confidence interval; F, female; BMI, body mass index; LDL, low-density lipoprotein; HDL-C, high-density lipoprotein cholesterol; NLR, neutrophil-to-lymphocyte ratio; MLR, monocyte-to-lymphocyte ratio; SIRI, systemic inflammation response index (neutrophil × monocyte/lymphocyte); Max-IMT, the maximal value among the measurements of the bilateral intima-media thickness of carotid artery; Stenosis%, maximum percentage stenosis of carotid artery.

**Table 4 jpm-12-01697-t004:** Multivariate linear logistic regression analysis for severity of carotid artery stenosis.

	B	SE	β	t	Sig.	VIF
SIRI	−0.782	0.681	−0.094	−1.147	0.253	1.089
Calcium kidney stone	9.574	2.641	0.295	3.624	<0.001	1.085
ASCVD risk score	6.715	1.837	0.287	3.655	<0.001	1.009

Abbreviations: B, regression coefficient, SE: standard error, SIRI: systemic inflammation response index, ASCVD: atherosclerotic cardiovascular disease.

## Data Availability

All data generated or analyzed during this study are included in this published article.

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
