# Peer review of "Risks of Carotid Artery Stenosis and Atherosclerotic Cardiovascular Disease in Patients with Calcium Kidney Stone: Assessment of Systemic Inflammatory Biomarkers"

_jpm, 2022, doi:10.3390/jpm12101697_

Round 1

Reviewer 1 Report

Dear Editor,

I have read the paper by Chan Jung Liu et al with interest. 

The authors have studied possible relationship of higher level of non-standard inflammatory markers (NLR, MLR or SIRI) with calcium kidney stones. Moreover, the authors conclude that the severity of carotid artery stenosis was higher among the calcium stone formers that non-stone formers.

Despite the paper is interesting I have found several minor shortcomings that should be mentioned. 

1. The protocol approval is referenced twice in "2.1. Study design and population." Please state it only once, with all the information.

2. Please include "gender" in the "Data collection" section for demographic variables.

3. Include the definition of systolic "BP". 

4. Correct line 79: participated -> participate.

5. Please include the abbreviations and symbol definitions from Table 4.

6. Is it possible to improve the aesthetic of Figure 2? The text is too narrow, and the styles on all figures should match.

7. I would suggest also implementing the discussion section. The authors should compare their data and NLR, MLR, and SIRI values with the data already published in literature regarding atherosclerosis and chronic disease, for example PAD (https://doi.org/10.3390/jcm11092620 , https://doi.org/10.3390/cells11071124 , https://doi.org/10.3390/jcm9061729 ), end stage kidney disease (https://doi.org/10.3390/life12091447 , https://doi.org/10.3390/biomedicines10061272 )

Reviewer 2 Report

The authors present interesting data showing the association between calcium kidney disease and higher levels of inflammatory biomarkers and carotid stenosis. In general, the manuscript is well written and structured and contains an important clinical message. In my opinion, this is a qualitative study that was an absolute pleasure to review. However, one fundamental omission must be addressed.

The sampling method presented by the authors appears to be a classic retrospective cohort study rather than a case-control study. This is because the sample of study participants was based on the presence of calcium nephrolithiasis, which was then observed toward the presence of the outcome. If the authors had selected participants based on the presence of carotid stenosis as an outcome (one group with the outcome and one group without the outcome) and examined the effects of urolithiasis in these two groups, this would be a case-control study in which the measure of association should be an odds ratio. In addition, to determine the proper study design, the time frame of the study should be specified. It is unclear when the data were collected: during treatment for urolithiasis/first examination of the control group or after patients were enrolled in the study.

Minor comment. I recommend that the authors present the figure of the ROC curve.
